# The Ecology of Subaerial Biofilms in Dry and Inhospitable Terrestrial Environments

**DOI:** 10.3390/microorganisms7100380

**Published:** 2019-09-23

**Authors:** Federica Villa, Francesca Cappitelli

**Affiliations:** Department of Food, Environmental and Nutritional Sciences, Università degli Studi di Milano, Via Celoria 2, 20133 Milano, Italy; francesca.cappitelli@unimi.it

**Keywords:** subaerial biofilms, inhospitable conditions, environmental stresses, mineral–air interface, lab-scale models, symbiotic playground

## Abstract

The ecological relationship between minerals and microorganisms arguably represents one of the most important associations in dry terrestrial environments, since it strongly influences major biochemical cycles and regulates the productivity and stability of the Earth’s food webs. Despite being inhospitable ecosystems, mineral substrata exposed to air harbor form complex and self-sustaining communities called subaerial biofilms (SABs). Using life on air-exposed minerals as a model and taking inspiration from the mechanisms of some microorganisms that have adapted to inhospitable conditions, we illustrate the ecology of SABs inhabiting natural and built environments. Finally, we advocate the need for the convergence between the experimental and theoretical approaches that might be used to characterize and simulate the development of SABs on mineral substrates and SABs’ broader impacts on the dry terrestrial environment.

## 1. Introduction

From the existence of extraterrestrial life in the universe to ancestral land colonization, from the drivers of primordial symbiosis to how to deal with antibiotic resistance, there are lots of phenomena we still largely do not know. Nevertheless, if at looked closely, all these big questions have a common denominator that is life in what humans consider inhospitable dry environments. With limited direct evidence, these issues remain scientifically problematic, but scientists can turn to indirect evidence to better address these questions. We think a good starting point would be the study of the adaption of microbial life (the principal biomass on Earth) to mineral substrates under environmental extremes, such as desiccation, radiation, high salt concentration, etc., which are features of dry terrestrial habitats.

The abiotic dry terrestrial environment is characterized by a patchy mosaic of air-exposed mineral substrates, ranging from natural (rocks) to man-made structures (e.g., stone heritage), subjected to a variety of environmental pressures and characterized by different physicochemical features. Although oligotrophic environments, mineral surfaces represent one of the main sources of micronutrients and ions potentially accessible to the biosphere [1]. Thus, it is not surprising to observe a thin veneer of densely packed microorganisms, called subaerial biofilms (SABs), that operate in self-organized structures on or within the top few centimeters of exposed soil and rocks [2]. In contrast to the planktonic mode of life, sessile aggregation provides cells with protection from various stresses including desiccation and antimicrobial agents and favors interplay among microorganisms and promotes social behavior through cooperation and the exchange of genetic material.

Interestingly, despite the inhospitable conditions in some habitats, the number of microorganisms in dry terrestrial environments is similar to the total number in marine habitats [3]. It also becoming increasingly clear that while the mineral–SAB interface is governed by microscale interactions played out within the mineralosphere, it exerts cascading bottom–up influences on ecosystem-scale processes such as primary production, the productivity and stability of food webs and biogeochemical cycling [1,4,5,6]. Thus, within the context of ecosystem function, the ecological relationships between minerals and SABs arguably represent some of the most important associations in dry terrestrial environments and land colonization, supporting a fundamental and seminal transition in microbial evolution [7].

Besides being nutrient-poor habitats, subaerial mineral substrates are often subjected to rapid changes in moisture availability, temperature and irradiation level, which promote the development of specialized SABs with efficient metabolic stress responses [8]. Thus, microorganisms within SABs must engage and coevolve with their neighbors and their physicochemical extracellular environment in order to adapt to diverse and fluctuating conditions.

Despite the abovementioned conditions and the low biomass, SAB communities are relatively stable [9] and display the community-level functional capability of maintaining a self-sustaining community [10]. The complex level of interspecies communication in SABs can be wholly appreciated by applying the current arsenal of omics-based technologies. Until recently, the study of SABs on mineral substrates has traditionally relied on the investigation of individually isolated microorganisms to make inferences about the entire community. However, our understanding of the SAB microbiota as an interactive ecosystem is still in its infancy. Although useful, it is now clear that studying communities through their individual components cannot adequately describe the collective behaviors and complex interplay that exist in multi-cellular communities under extreme conditions. Indeed, most of the time, there is a syntrophic chain in which several species contribute to the transformation of a compound [11].

Using life on air-exposed minerals as a model and taking inspiration from the mechanisms by which some microorganisms have adapted to the most inhospitable conditions, we applied “systems thinking” to illustrate the ecology of SABs inhabiting natural and built (stone heritage) environments. Finally, we advocate the need for the convergence between the experimental and theoretical approaches that might be used to characterize and model the development of SABs on mineral substrates and their broader impacts on the dry terrestrial environment.

Overall, we argue that solutions to environmental stresses would greatly benefit the knowledge we can gather from the ecology of SABs.

## 2. Community Assembly at the Mineral-Air Interface

Although one might assume that the dry terrestrial environment is generally inhospitable, it in fact hosts a large diversity of microbial life that preferentially accumulates along crevices and fissures, and the cleavage steps and edges of mineral surfaces [12,13,14].

Given the stark and oligotrophic nature of mineral substrates in many dry terrestrial environments, it is not surprising that SABs are supported by photosynthesis-based interactions (Figure 1) [2]. Carbon fixation by phototrophs drives chemoorganotrophic assemblages in this ecosystem. This points to the fundamental role of the mineralosphere as a key meeting place for shaping phototroph–chemoorganotroph partnerships [15]. The main characteristics of SABs inhabiting natural and built (stone heritage) environments are summarized in Table 1. It is possible to draw parallels among key biofilm attributes that drive geomicrobial processes in the topsoil, rock and man-made structures in an inhospitable environment by comparing the main phylogenetic groups, functional traits, biologically-driven processes and drought-resistance mechanisms, as reported in Table 1.

The next-generation sequencing of SABs on mineral substrates is becoming increasingly available, which has enabled a more detailed understanding of the composition and diversity patterns of microbial communities within these ecosystems. Although SABs inhabiting desert rocks and stone heritage can be highly specific to their lithic substrate, they were all dominated by the same four main phyla, Cyanobacteria, Actinobacteria, Chloroflexi and Proteobacteria [16,33,34,37,38,39]. While the phylum Ascomycota is the most representative of the fungal communities, the archaeal community is the least represented in SABs on mineral surfaces.

Overall, the results revealed taxa associated with survival under extreme conditions such as salinity (e.g., *Halococcus*, *Kocuria*, *Salinimicrobium*, *Pontibacter*, *Halobacterium*, *Marinobacter*, and *Halomarina*), UV radiation (e.g., *Thuepera*, *Deinococcus*, *Coccomyxa*, *Rubrobacter*, *Chroococcidiopsis*, *Spirosoma*, *Scytonema*, *Blastococcus*, and *Modestobacter*), acidic conditions (e.g., *Apatococcus*, *Acidobacteriaceae*, *Beijerinciaceae* and *Methylocystaceae*), alkaline conditions (e.g., *Spirosoma*, *Rubellimicrobium* and *Truepera*) and low water availability (e.g., *Chroococcidiopsis*, *Knufia*, *Leptolyngbya*, *Sarcinomyces*, *Rubrobacter*, *Capnobotryella*, and *Scytonema*) [17,35,40,41,42]. Other genera indicated the capacity to cycle nitrogen (e.g., *Thiobacillus*, *Malikia*, *Ochrobactrum*, *Nitrososphaera*, *Nitrospira*, *Novosphingobium*, and *Nitrobacter*), cycle sulfur (e.g., *Thioclava*, *Thiobacillus*, *Rhodovulum*, and *Desulfuromonas*), autotrophically fix carbon (e.g., *Chroococcidiopsis*, *Leptolyngbya*, *Nostoc*, *Trebouxiophyceae*, *Phormidium*, *Aurantiamonas*, *Thiobacillus*, *Thioclava*, *Rhodobacter*, and *Acidimicrobium*), utilize minerals such as iron and manganese (e.g., *Aurantimonas*, *Acidimicrobium*, and *Ferrimicrobium*) and bioprecipitate minerals (e.g., *Bacillus*, *Stenotrophomonas*, *Pseudomonas*, and *Crosiella*) [10,29,30,35,42].

Despite the sequencing efforts and advanced analytical tools, the complete genomes of microbial species from the topsoil, rock and man-made structures are scarce (Table 2). Genome sequences from cultivated and uncultivated microorganisms will allow deep investigations of the physiological traits that enable survival under inhospitable conditions, including the ability of these microorganisms to respond to future perturbations such as climate change and human impacts.

Furthermore, the scientific community is still far from understanding the mechanisms behind the formation of SAB communities on mineral substrates. How do SAB communities assemble? What are the main drivers for structuring SAB communities on minerals?

By sampling tombstones across three continents, Brewer and Fierer [36] demonstrated that the type of stone had a major effect on the overall composition of the SAB communities. In line with this finding, shot gun metagenomics analyses identified gene categories that were differentially abundant across two lithotypes, granite and limestone [36]. Granite samples harbored acidophilic bacteria and showed gene pathways linked to both acid resistance and acid production, cell movement and substrate transport as well as amino acid synthesis. By contrast, limestone surfaces hosted radiation-resistant bacteria with neutral to alkaline pH growth ranges as well as lichen-associated taxa. Limestone communities were enriched in genes related to carbon-fixing pathways, UV resistance and vitamin and cofactor synthesis.

Li and colleagues [9] studied the epilithic SABs of archeological sites located at different areas of China over a two-year period. They investigated whether biofilm communities, which were geographically and chronologically separated, showed distinct taxonomy or functionality. Findings indicated that no substantial differences in terms of community structures were observed among the different locations, while microhabitat was the major factor affecting the stone microbiome. Functional prediction analyses indicated that the ATP-binding cassette transporter (ABC transporter) system was characteristic of the deterioration-associated microbiome, indicating that the complex exchange of amino acid, molecules and iron complex is affected [9]. Furthermore, samples showing plaques of white deposits on the surface were characterized by metabolic pathways involving mineral absorption, calcium signaling, extracellular polymeric substance (EPS) production and membrane transport proteins, suggesting the occurrence of CaCO_3_ precipitation processes [9].

A recent study carried out on SABs colonizing the passage of Lascaux Cave revealed that mineral substrates were important drivers structuring both the total (DNA) and the metabolically active (RNA) microbial communities, more so than the presence of black discoloration and the seasonality did [43].

Substrate-dependent patterns of community assembly have been observed in biological soil crusts and cryptoendolith communities from the Colorado Plateau Desert, as well as in hypolithic communities colonizing the meteorites found in Nullarbor Plain, Australia [44,45]. By studying the diversity and community composition of endoliths from four different lithic substrates collected in the Atacama Desert, Meslier et al. [46] showed how SAB assembly was driven by substrate properties. Among the substrate properties, rock architecture and water retention capabilities are the main factors influencing microbial community compositions.

Altogether these findings suggest that distinct mineral substrates from the same climate regime and geographic area harbor distinct microbial communities.

**Table 2 microorganisms-07-00380-t002:** Complete genomes of microbial species from the topsoil, rock and man-made structures.

	Sequenced Genomes
**Biological Soil Crusts**	*Microvirga* sp. Strain BSC39 [47] *Aquincola tertiaricarbonis* [48] *Microcoleus vaginatus* FGP-2 [49] *Massilia* sp. Strain BSC265 [50] *Bacillus* sp. Strain BSC154 [51]
**Desert Rocks**	*Knufia petricola* [52] *Sphingomonas* sp. strain AntH11 [53] *Rachicladosporium antarcticum* CCFEE 5527 and *Rachicladosporium* sp. CCFEE 5018 [54] *Halorubrum* sp. SAH-A6 [55] *Nakamurella lactea* [56] *Cryomyces antarcticus* [57]
**Stone Surfaces**	*Hassallia byssoidea* strain VB512170 [58] *Scytonema millei* VB511283 [59] *Tolypothrix boutellei* strain VB521301 [60] *Blastococcus saxobsidens* DD2 [61] *Modestobacter marinus* strain BC501 [62]

## 3. Biological Interactions in SABs: A Symbiotic Playground

SABs have been the focus of numerous studies on microbial community structure and function, improving our understanding of their relative trophic simplicity and pertinence in ecosystem service maintenance [16,40,63].

Metagenomic studies of SABs have revealed stress response and nutrient cycling genes to fix carbon and nitrogen under fluctuating and inhospitable conditions, as well as genes involved in microbial competition and cooperation [64,65,66].

Lichens, associations of a fungus and a chlorophyll-containing partner (either green algae or cyanobacteria, or both), represent iconic examples of symbiosis at the mineral-air interface. Thanks to this association, the production of energy via carbon dioxide fixation provided by photobiont is enhanced by the sheltering structures offered by the fungal partner. However, this classical view has been challenged by recent omics investigations, which revealed a functional contribution of bacteria associated with lichens, corroborating the understanding of lichens as stratified biofilms [67] and characterized by multi-species symbiosis [68,69]. Grube et al. [70] provided strong evidence that the bacterial microbiome is involved in multiple aspects of the symbiotic system, including (i) nutrient provision, especially nitrogen, phosphorous and sulfur, (ii) resistance against biotic stress factors (iii) resistance against abiotic factors, (iv) biosynthesis of vitamins and hormones, (v) detoxification of metabolites, and (vi) degradation of older lichen thallus parts. Furthermore, recently, integrated metaproteomics analyses showed that lichen symbionts are involved in specific activities. Fungal symbionts produce transport protein-regulating vesicle traffic, cyanobacteria synthetize nitrogenase and glutamine oxoglutarate aminotransferase involved in nitrogen fixation, algae express proteins functioning in photosynthesis, and bacterial enzymes are responsible for methanol/C1-compound metabolism as well as CO-detoxification [71].

Within SABs, mutualistic interactions between autotrophs and heterotrophs can occur beside lichen symbiosis. Geological evidence of interspecies interactions between cyanobacteria and heterotrophs dates to a million years ago, and the cohesiveness of such interactions is demonstrated by the paucity of axenic cyanobacteria strains [72]. More recently, Couradeau et al. [73] described the mutualistic relationship between the dominant member of the biocrust microbiome, the non-nitrogen-fixing cyanobacterium *Microcoleus vaginatus*, and the diazotrophic copiotrophic heterotrophs based on a C for N exchange. In fact, while *M. vaginatus* offers organic carbon, it relies on other bacteria for its nitrogen needs. Thus, *M. vaginatus* carries its own built-in nitrogen fixation “microbiome module” in order to increase its fitness as a colonizer of N-depleted soils [74].

A study carried out by Villa and colleagues [11] explained cooperative interactions in SAB communities by revealing the functional interplay occurring among four main microbial groups inhabiting a tombstone located in a polluted environment. It was observed that the organic carbon produced by cyanobacteria during photosynthesis fuels sulfate-reducing bacteria (SRB) and sulfur-oxidizing bacteria (SOB) growth, while the photosynthetic oxygen is consumed by SOB, generating the anaerobic environment for SRB and anoxygenic phototrophic sulfur bacteria. Furthermore, SOB activity quickly removes S^2−^, the metabolic product of SRB, which could inhibit cyanobacteria and, at higher concentrations, also SRB. Genes for assimilatory sulfate reduction, mineralization of organic sulfur compounds, and oxidation of sulphide and thiosulphate were detected in the metagenome of hypolithic SABs from the Namib desert, suggesting an extensive capacity for sulfur cycling [65]. Zanardini et al. [10] demonstrated that SABs on stonework are potentially self-sustaining ecosystems able to cycle essential elements such as carbon, nitrogen and sulfur. Through functional gene analyses, the researchers found a complete nitrogen cycle with nitrogen-fixing cyanobacteria, nitrifying and denitrifying bacteria as well as the presence of autotrophic carbon fixation capacity and sulfur-oxidizing bacteria and sulfate-reducing bacteria.

However, the structurally and functional complexity of SAB communities make species-specific observations of behavior technically challenging. The use of synthetic consortia maintained under controlled environments is attractive to infer mechanisms that mediate symbiotic relationships such as metabolic coupling and acclimation to partnership.

The unicellular cyanobacterium *Synechocystis* sp. strain PCC 6803 and the chemoheterotroph *Escherichia coli* K12 were used to reproduce a laboratory-scale dual species SAB on limestone [74]. Findings demonstrated that cyanobacterial biomass and stress resistance increased as a result of heterotrophic partnership. Furthermore, the cyanobacterial matrix offered carbon and energy sources for *E. coli* growth, while the heterotroph promoted cyanobacteria growth by providing key metabolites and the scavenging of waste products [72,74,75]. The involvement of cooperative behavior may suggest the participation of quorum sensing signaling molecules in such responses. Quorum sensing (QS) is a cell-to-cell communication system depending on population density able to coordinate community behavior. Quorum sensing involves production of and response to diffusible or secreted signals called autoinducers. Sharif et al. [76] reported that the epilithic cyanobacterium *Gloeothece* produces N-octanoyl-homoserine lactone as a signal molecule, altering gene expression in response in an autoinducer-like manner. Through microscopic investigations, Villa et al. [2,74] revealed that the EPS matrix was not concentrated in only one single colony but extended along the mineral substrata, interconnecting cellular clusters together. Interestingly, it has been observed that the “calling distance” of quorum sensing can extend up to 78 μm between single species biofilms [77]. Thus, it is likely that communication and cooperation among segregated microcolonies can occur by diffusion of metabolites and QS molecules through the EPS network.

Species-resolved transcriptomic analyses of binary consortia composed by a cyanobacterium and an obligate aerobic heterotroph revealed that the phototroph responded to the heterotrophic partnership by altering the expression of core genes involved in photosynthesis, carbon uptake/fixation, vitamin synthesis, ribosomal proteins, and scavenging of reactive oxygen species [78,79,80,81]. El Moustaid et al. [22] studied how photosynthesis can be conflicted by photorespiration in a dual species consortium composed of a phototroph and a heterotroph. Results showed that a phototroph-heterotroph consortium can increase biomass by recycling photorespiration byproducts.

Overall, the above results suggest that SAB communities prefer mutually neutral or even beneficial associations, expanding metabolic abilities and improving resource utilization and stress responses over that of its individual members [8]. That is why SABs have been considered as symbiotic playgrounds [8].

## 4. Stress Resistance and Resilience of SABs

SABs are likely equipped to cope with frequent, often daily, stress factors such as hydration/dehydration cycles, extreme fluctuations in temperature and irradiance as well as biocide treatments. Understanding the mechanisms behind stress responses in SABs is instrumental to identify the ecological and physiological drivers of biofilm formation, resistance (insensitivity to disturbance) and resilience (the rate of recovery after disturbance) in a changing environment. So far, little is understood at community level, while several studies are available for single isolated terrestrial species.

### 4.1. Physical Stresses

SABs show a pronounced three-dimensional stratification, where, in most cases, phototrophs dominate closer to the biofilm-air interface in contact with the external environment [2]. Thus, studying self-protection strategies of phototrophs is important as the vertically lower-positioned microorganisms of a SAB may not have ever been exposed to some physical stresses thanks to the efficient response of these specialized microorganisms on the top of the biofilm [25]. Photosynthesis is affected by drought and UV-B irradiation. Studies on *Leptolyngbya ohadii* as well as *Microcoleus* sp. isolated from BSCs showed the activation of a nonradioactive cyclic electron transfer route during photosynthesis that minimized oxidative damage in desiccation-tolerant cyanobacteria [82,83]. During the desiccation of *L. ohadii*, a decrease in the transcript levels of genes involved in light harvesting, photosynthetic metabolism, protein biosynthesis and cell division was detected [84]. *Phormidium tenue* responded to UV-B radiation reducing photosynthetic activity, while increasing the production of antioxidant enzymes, DNA damage repair systems, and UV-absorbing pigments to protect itself from the cell damage caused by the radiation [85]. Recently, Wadsworth et al. [86] investigated the survival of *Gloeocapsa* sp. in extraterrestrial conditions. The findings demonstrated that metabolically inactive cells surrounded viable cells, providing protection against environmental stress such as UV radiation. Thus, viable cells can bury themselves in disordered aggregates of sheathed inactive cells, which also provided a protected niche for other bacteria that survived in space [86].

As expansion and shrinkage could cause mechanical stress upon desiccation, a β-galactosidase produced by the terrestrial cyanobacterium *Nostoc flagelliforme*, the transcription of which is regulated by moisture cycles, has been demonstrated to affect EPS density [87].

The amount of proteins has been shown to be modulated in response to stresses although both a decrease and an increase have been claimed. Steven et al. [18] observed an increased protein production from the same number of mRNA molecules, while [32] microcolonial fungi (MCF) exposed to desiccation/rehydration events, demonstrated a loss of proteins. Wang et al. [88] performed a comparative transcriptome analysis of the lichen-forming fungus *Endocarpon pusillum* to elucidate its drought response and found an up-regulation of genes coping with proteins misfolding. The researchers also observed the differential expression of genes involved in sugar synthesis, suggesting that under desiccation the mycobiont reduced monosaccharides and increased polysaccharides.

Murik et al. [89] also suggested that the synthesis and degradation of various osmolytes are used to balance the changing water potential in *L. ohadii*. Interestingly, nitrification in the enrichments of some Negev desert samples was performed up to 400 mM NaCl (2.3% salt), a concentration close to that of seawater [90]. It is known that salt accumulation increases extracellular osmolarity as desiccation.

### 4.2. Chemical Stresses

Zhang et al. [91] investigated the importance of salinity along a natural salinity gradient in the Gurbantunggut Desert, Northwestern China. The researchers proved that microbial diversity was linearly reduced with salt accumulation, but community dissimilarity greatly increased with salinity differences. The latter counterintuitive finding has been explained by the fact that unrelated taxa also coexist in dry environments and the competitive exclusion of some closely related taxa.

Despite some manuscripts on salinity, the resistance and resilience of SABs to chemical stressors have been focused on the use of biocides in conservation treatments of stone monuments [92]. However, the main question we pose is how can we tell whether a biocide treatment is working against SABs?

In this respect, Villa and co-workers [74] investigated the susceptibility of a dual-species SAB to the quaternary ammonium solution D/2 through time lapse confocal laser scanning microscopy. By quantifying the fluorescence loss of both green fluorescence protein (GFP)-tagged heterotrophic cells and autofluorescent phototrophic cells, this technique permitted the direct visualization of cell inactivation patterns in biofilm structure during biocide action (Figure 2, Appendix A). The extent of fluorescence loss ranged from 46% to 80% depending on the bacterial group considered, suggesting an intrinsic resistance to traditional biocidal active substances and a different level of susceptibility towards the chemical compound applied.

Another question addresses which microorganisms the selected biocides are efficiently targeting. Urzì et al. [93] compared the change occurring to the SAB communities in one site of the Catacombs of Domitilla before and after a treatment with a mixture of quaternary ammonium compounds and actylisothiazolone applied for a one-month period. The results clearly showed that cyanobacteria were slightly affected by the treatment, while the heterotrophic bacteria changed drastically in terms of diversity. In fact, the synthesis and excretion of extracellular polymeric substances, mainly produced by cyanobacteria, can provide a barrier against the penetration of biocidal compounds [94,95]. Nowicka-Krawczyk and colleagues [96] investigated the effect of silver nanoparticles (AgNPs) on the most frequently occurring species of green algae in subaerial biofilms. Overall, their findings demonstrated that although all the tested AgNP concentrations affected the growth of the aerophytic algae in a dose-dependent manner (with a biomass reduction ranging from 26% to 68%), the inhibition was time dependent and, in some cases, it was reversed after two weeks from the treatment. Gambino et al. [97] exposed fungal biofilms, grown with a colony biofilm approach, to two concentrations (0.25% and 0.5%) of zinc oxide NPs (ZnO-NPs) for 10 days. It was shown that the growth rate of some fungal biofilms at both 0.25% and 0.5% concentrations was severely slowed down by the ZnO-NPs, while the treatments did not successfully work for *Aspergillus niger* biofilms, its growth being promoted by the lower ZnO-NPs concentration.

In addition, SAB communities may become less sensitive or even resistant to the biocides, exerting a harmful impact to the object. The Lascaux Caves is an emblematic example of situations in which a series of biocide treatments over time (antibiotics, formol, various products based on benzalkonium chloride and isothiazolinone) triggered the development of white patina caused by *Fusarium solani*, the growth of resistant *Pseudomonas fluorescens* strains and the growth of melanized fungi such as *Ochroconis lascauxensis*, *Ochroconis anomala* and *Exophiala castellanii* [98].

The resistance of SABs is clearly not caused by a single factor, but rather by several mechanisms acting in concert, ensuring the survival of biofilm cells in the face of even the most aggressive antimicrobial treatment regimens. The mechanisms include, among others, poor biocide penetration, nutrient limitation and slow growth, adaptive stress responses, the formation of persister cells and various actions of specific genetic determinants of biocide resistance and tolerance [99].

EPSs are polymers that can interact with biocides by inactivating the molecule through delaying their diffusion or chemically react with them [100]. Furthermore, in SABs, the EPS matrix provides heterogeneous nucleation sites for calcium carbonate precipitations [101]. It has been observed that many photosynthetic microorganisms inside SAB communities can bury themselves under calcite crystals, which results in an extra barrier against the biocidal action [93].

Taken together, the results suggest the pervasiveness and diversity of resistance in non-pathogenic bacteria, such as those retrieved in SABs, despite the lack of human intervention. The reservoir of resistance elements might be related to the resistance and resilience of SABs to antimicrobial agents.

## 5. Lab-Scale Systems and Mathematical Models: Methods to Study SABs

Simplified laboratory-based model systems become instrumental for exploring SAB performance and for the application of omics-based approaches to dissect subcellular pathways and regulatory networks. Furthermore, the simplicity and controllability of model systems provides a stark contrast to the complexity and inaccessibility of field systems. The model systems enable the researchers to test hypotheses about the ecology of SABs under different environmental conditions, and to establish the plausibility of mechanisms governing biogeochemical transformations [2].

Despite the success in reproducing complex phototrophic SABs [102,103,104], these communities are difficult to analyze at the molecular level since omics technologies require microorganisms with available genetic and physiological information [105]. To overcome the limitation offered by complex phototrophic SABs, an elegant dual-species model biofilm was developed. The model system comprised the cyanobacterium *Nostoc punctiforme* strain ATCC 29,133 (PCC 73102) as a phototroph, and the well-studied marble-derived isolated microcolonial fungus A95 *Knufia petricola* (syn. *Sarcinomyces petricola*) [106] as a heterotrophic component [8]. Through this model, Seiffert and colleagues [107,108] successfully studied the biological impact of the consortium on weathering granite and related minerals in a new setting of a geomicrobiologically modified percolation column. Villa et al. [74] proposed a laboratory model of SABs composed of the unicellular cyanobacterium *Synechocystis* sp. strain PCC 6803 and the chemoheterotroph *E. coli* K12 (Figure 3). Villa and colleagues’ model system was able to mirror the main features of biofilms inhabiting lithic substrates, such as morphology, syntrophic interactions, survival to desiccation stress and biocide tolerance. 

Despite the importance of SAB models, quantitative studies using mathematical or computational approaches are rare. In this respect, through a model, it has been shown that under periodic stress, *Pseudomonas aeruginosa* planktonic cells preferentially attach to existing aggregates rather than to a bare surface. In inhospitable environments, this phenomenon can favor a mixed biofilm development [109].

The first work concerning a conceptual model of rock dwelling fungal biofilms formed on exposed surfaces of solid rocks was proposed by Chertov et al. [110]. By simulating the growth of a single microcolony on a rock surface, the researchers proved whether the fungal growth is influenced by the environmental factors and organic compounds. Process-based models have also been developed for biological soil crusts of lichens and mosses, to predict processes that control carbon uptake (photosynthesis, respiration, water uptake and evaporation), global biogeochemical cycles and weathering [19,111,112,113,114]. Recently, Kim and Or [63] reported a mechanistic model that considers the physical and chemical processes shaping the functioning of biocrust communities that interact and respond to cycles of hydration, light, and temperature. The model results showed not only the distribution and composition of microbial functional groups over vertical gradients of light, temperature, and substrates, but also carbon and nitrogen cycling within the biocrust. Furthermore, the findings based on an acid-base equilibrium predicted the spatial and temporal activity of microbial functional groups. Self-organization explains why biocrusts can host high biodiversity even under very dry conditions like deserts.

## 6. Conclusions and Future Directions

The next-generation sequencing of SABs on mineral substrates is becoming increasingly available and shows the dominance of four main phyla, Cyanobacteria, Actinobacteria, Chloroflexi and Proteobacteria. Thus, SAB communities living in mineral environments benefit from goods exchange and labor division in long-term partnerships between members of different groups.

Subaerial biofilms in inhospitable environments are multi-component open ecosystems sensitively tuned to the external environment, which provides not only nutrients, moisture and space, but also physical and chemical stressors that are drivers of biofilm formation, resistance and resilience. Despite the significance of model systems in SAB studies, works on this topic are rare, and most of them are based on mono-species biofilms.

Despite the ubiquity of SABs and their effects over natural and man-made ecosystems, there are still many unanswered questions regarding their physiology and behavior as ecological systems. Future research should be aimed at opening up the ‘black box’ of SAB biomes for unravelling the assembly of microbial communities (diversity) and their response to extreme conditions in a spatio-temporal scale (function) in line with the extracellular physical and chemical environment. For instance, the dispersal and invasion of SAB microorganisms across local, regional and continental scales are poorly investigated, although they are a major factor in shaping the biogeography of SABs. Unravelling the biogeographic patterns of SABs could help in improving large-scale ecosystem models of greenhouse gas fluxes and the response of these fluxes to further changes in the global climate and atmospheric composition.

At present, stress response mechanisms at community assembly levels are neglected but they have the potential to reveal novel adaptive traits in SAB microorganisms and a better understanding of the colonization processes of mineral substrates under stress conditions. This understanding will help to reduce uncertainties about the responses of microorganisms to a change in environment and will enable that knowledge to be incorporated into future predictive models of climate change and some terrestrial feedbacks.

It will also be intriguing to establish to what extent biogeological processes and interactions with the environmental conditions can be predicted by the characteristic of SAB communities (either the composition or functional markers), and whether a unifying theory can be developed to explain the role of SAB in many different inhospitable dry terrestrial ecosystems. Thus, understanding interactions among SABs, mineral substrates and the changing environment is crucial to addressing ecological and biogeochemical questions, as well as to developing tools for predicting what their potential is to influence climate feedbacks across ecosystems and along environmental gradients.

Despite advances in our knowledge of the physiology of SABs, we are still far from achieving the level of fundamental understanding of their dynamics and functions that is needed to predict and manipulate SAB behavior in dry terrestrial environments. SAB dynamics and functions could be very important, since SABs could either participate in climate change mitigation or exacerbate anthropogenic climate change. Omics-based approaches allow us to catalogue the structure and function of SABs to an unprecedented level of detail. Omics data, along with environmental and substrate information, represent a snapshot of the SAB ecosystem. The key challenge now is to incorporate biological (omics), environmental, chemical and geological data into mathematical models, in order to offer a system-level understanding of the phenomenon. To achieve this level of predictive understanding, integration between mathematical models, with a basis in mechanistic understanding, and controlled experiments is required. Thus, increased interaction between empiricists and theoreticians, as well as the development of standardized SAB models, which can act as testbeds for the development of experimental and modeling approaches, is strongly encouraged.

Understanding how SABs cope with extreme conditions can help elucidate the potential for life to exist in a context of environmental changes and even beyond our planet.

## Figures and Tables

**Figure 1 microorganisms-07-00380-f001:**
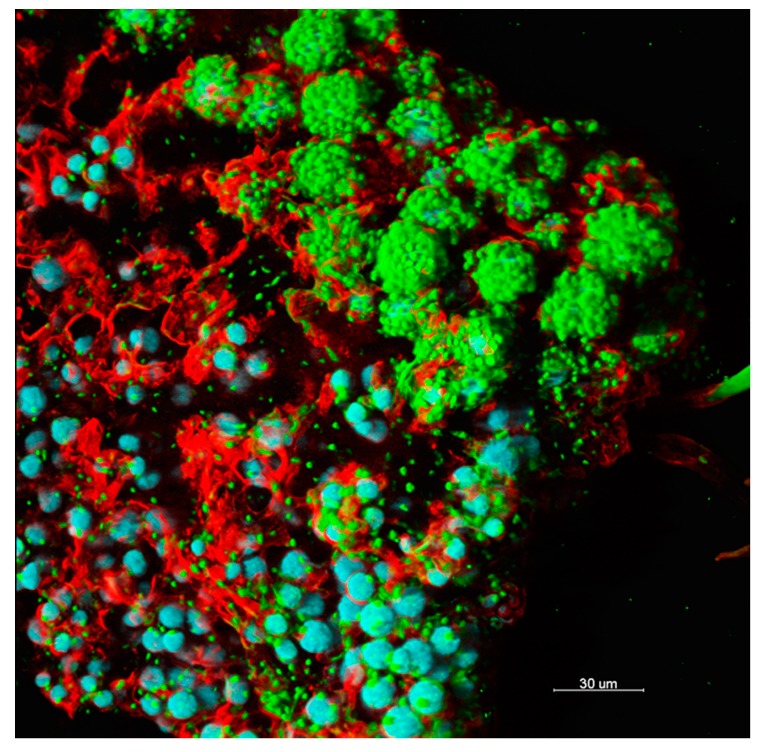
Confocal laser scanning imaging of a subaerial biofilm (SAB) growing on a stone monument. Blue, microcolonies of photoautotrophic microbes; green, chemotrophs; red, extracellular polymeric substances.

**Figure 2 microorganisms-07-00380-f002:**
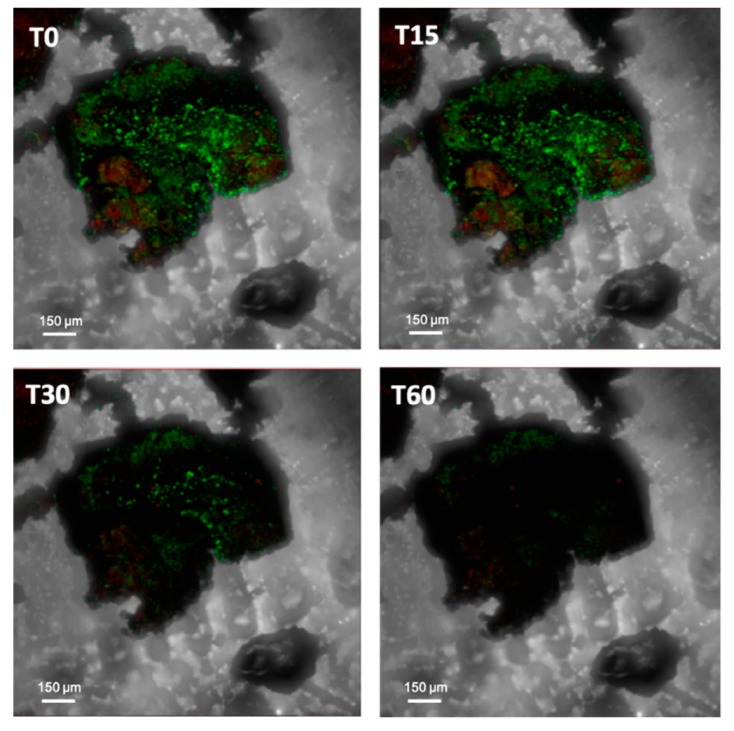
Antimicrobial effectiveness of D/2 solution. Real-time loss in cell viability over time in the presence of the biocide treatment (treated sample). Chemotrophs have been visualized in green by Calcein AM, which detects metabolically active cells. Phototrophs have been visualized in red by using the natural autofluorescence of the photosynthetic pigments. It is possible to observe the presence of green and red signals after the treatment, indicating cells resistant to the biocide.

**Figure 3 microorganisms-07-00380-f003:**
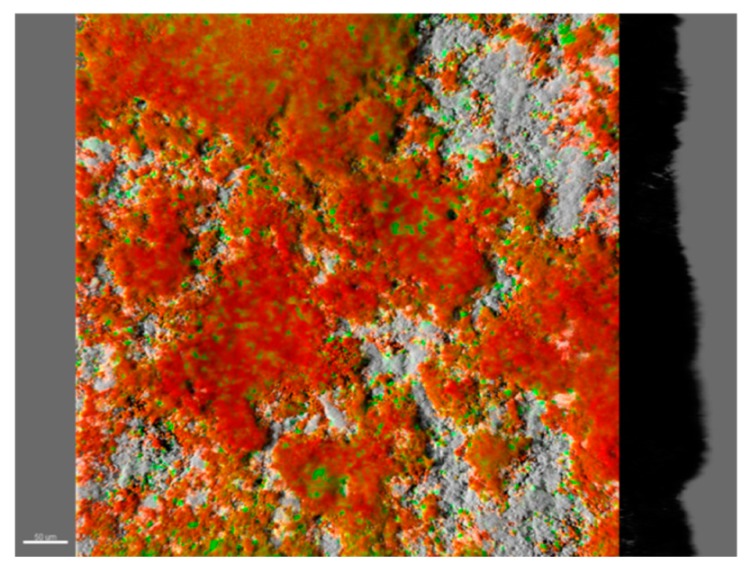
The 3D reconstruction of cell aggregates in a dual-species SAB model system. Color key: *E. coli* cells, green (green fluorescence protein (GFP)); *Synechocystis* cells, red (autofluorescence); reflection from inorganic materials, grey.

**Table 1 microorganisms-07-00380-t001:** Main characteristics of SABs as biological soil crusts or inhabiting natural (desert rocks) and built (stone heritage) environments.

	Core Microbiome (Metagenomic Studies)	Main Functional Traits	Biologically-Driven Processes	Mechanisms of Drought Resistance
**Biological Soil Crusts**	Bacteria: Cyanobacteria, Actinobacteria, Acidobacteria, Alpha-proteobacteria, and Bacteroidetes Fungi: Ascomycota, Basidiomycota, and Chytridiomycota Archaea: Crenarchaeota [16,17,18,19,20,21]	Functional genes associated with C degradation and N cycling. [20]	Modulating C and N heterogeneity and cycling. Increasing the capture of nutrient-rich dust. Modulating, surface albedo, water fluxes and erosion. Influencing soil fertility and plant establishment patterns. [22,23,24]	EPSs act as a repository for water and stabilize desiccation-tolerant enzymes and molecules. Activation of a non-radioactive cyclic electron transfer route during photosinthesis to minimize oxidative damage. Synthesis and degradation of osmolytes are used to balance the changing water potential. [23,24,25]
**Desert Rocks**	Bacteria: Actinobacteria, Cyanobacteria, Proteobacteria, and Chloroflexi Fungi: Ascomycota Archaea: Crenarchaeota [26,27,28]	Transition metal-related molecular functions such as manganese ion binding and iron ion binding. [29,30]	Modulating C and N heterogeneity and cycling. Clogging the surface rock pores through secretion of extracellular polymeric substances (EPSs), lowering evaporation and slowing salt crystallization. [31]	EPSs act as a repository for water and stabilize desiccation-tolerant enzymes and molecules. Synthesis of heat-shock proteins and chaperons. Production of antioxidant enzymes, DNA damage repair systems, and UV-absorbing pigments. Dormant cells. [23,32]
**Stone Heritage**	Bacteria: Cyanobacteria, Actinobacteria Proteobacteria, Bacteroidetes, Acidobacteria, and Chloroflexi Fungi: Ascomycota Archaea: Euryarchaeota and Crenarchaeota [33,34,35,36]	Functional genes associated with C, N and S cycling autotrophic carbon fixation and mineral transformation processes. [10,11]	Modulating C and N heterogeneity and cycling. Weakening of the mineral lattice through wetting and drying cycles and sub-sequent expansion and contraction of the EPS matrix. Dissolving minerals through the excretion of H^+^, CO_2_, organic and inorganic acids, siderophores and other metabolites. Mediating the formation of minerals. Regulating water fluxes in the stone. Increasing hydrophobicity of the surface. Incorporation of mineral grains into the biofilm. [14]	EPSs act as a repository for water and stabilize desiccation-tolerant enzymes and molecules. Synthesis of antioxidant. Synthesis and degradation of osmolytes to balance the changing water potential. Dormant cells. [23,32]

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
