# Peer review of "The Ecology of Subaerial Biofilms in Dry and Inhospitable Terrestrial Environments"

_microorganisms, 2019, doi:10.3390/microorganisms7100380_

Round 1

Reviewer 1 Report

Comments on the manuscript of a review paper entitled “A closer look at mineral substrates: the ecology of subaerial biofilms in harsh terrestrial environments” submitted to Microorganisms by Federica Villa and Francesca Cappitelli

Summary:

This manuscript aims to review capabilities and survival mechanisms of subaerial biofilms in varying environmental conditions and stresses. First the effect of some mineral substrates on microbial colonisation are discussed, followed by a review of symbiotic communities in subaerial biofilms, responses to drought, radiation and chemical stress, and finally experimental and analytical methods. The focus is on (temporarily) dry conditions and UV radiation tolerance, although this is not very clearly stated in the manuscript. Instead, very broad definitions like “harsh”, “extreme” and “terrestrial” are, often misleadingly, used. The rationale and aims of the review are valid, and recent literature is cited. However, the ambitious goals set in the abstract and introduction are not met. In many places the manuscript seems to be more like a list of summaries of previous studies rather than a proper review, which should compare, combine and make new connections between and conclusions from the existing data and studies. I suggest that the authors make the focus clearer and change the structure of the text such that the content and interpretations rather than details of individual studies are highlighted. Limitations of the methods and critical evaluation of the cited literature should also be included. More detailed comments are provided below.

Major comments:

The title of the manuscript is misleading. Properties of mineral substrates are not discussed much, but instead the focus is on microbial communities. On the other hand, the latter part of the title alone is too broad considering the focus of the review. Please revise the title such that it describes the content of the manuscript.

In the abstract extreme and harsh conditions are mentioned without explanation. Please avoid too general terms and replace them by better defined terminology or explain what is meant here by extreme and harsh. Furthermore, I don’t see how climate change, extraterrestrial environments or system thinking mentioned in the abstract are considered and covered in the text, except in few separate sentences. Please make sure that the same topics are addressed in the abstract and the main text.

There are many parts of the manuscript where specific terminology is used either loosely or without explanation. For example, terrestrial environments and lithosphere cover a wide range of different environments from deserts (both warm and cold) to aquifers (e.g. lines 82 and 86), lithosphere even including the oceanic crust. Thus, not all terrestrial or lithospheric environments are subaerial, dry or inhospitable. Again, better define and keep the focus and use the terminology accordingly. For the sake of clarity, please also give short explanations for some specific terms such as chaperone gene and WspA (what are they doing, why are they important in this context?).

Especially in chapters 4 and 5 the review turns into a list of previous studies. The text could be condensed and it should report similarities and possible contrasts rather than detailed results of individual studies. This is also reflected in the conclusions section of the manuscript, in which conclusions are not given, but the focus is solely on future directions/lack of data.

Detailed and minor comments:

Line 12: Minerals substrates alone cannot be considered as ecosystems. Please be more specific.

Lines 13-14: What is meant here by extreme conditions?

Line 17: Harshest conditions is a strong statement. What about microbes living in hot springs or inside nuclear reactors? Again, please be more specific what you mean by harsh.

Line 39: Mineral surfaces alone cannot be considered as specific environments, as water content, amount of organic matter and exposure to air can vary greatly between different environments containing mineral surfaces.

Line 44: Replace comma with “and” between desiccation and antimicrobial.

Lines 46-47: Again the term harsh is used loosely. Also some marine habitats can be considered harsh. What part of the terrestrial biomass/biofilms is subaerial, i.e. more relevant to this review (also see the line 60, where low biomass is mentioned)?

Line 55: Add “subaerial” before mineral substrates.       

Lines 69-70: Please add reference(s) to the sentence “Indeed, most of the time…”

Lines 70-74: Consider moving this paragraph to section 2.

Lines 86-88: Please add reference(s), and also more clearly focus on subaerial biofilms; lithosphere is way too wide term to be used here.

Table 1: Layout of the table makes it difficult to read. For example it is not clear what references belong to what content.

Line 109: Replace “latest” by “least”.

Lines 140-145: Reference(s) for this paragraph are missing.

Section 3.: Please consider reorganizing this section. It would be more logical to start with the classical example of lichens (from line 192 on) and only after that continue to the more recent examples and ideas. The section can also be shortened by condensing some of the more detailed content (same applies to section 4).

Lines 219-220: This sentence should be rephrased, or maybe move “technically challenging” to the end?

Lines 239-240: Please add reference(s).

Lines 244-246: This paragraph (probably from a previous review report?) should be removed.

Line 257: What studies? Please avoid using these, this etc. especially in the beginning of a paragraph.

Line 266: Abbreviation EPS is used for the first time, please explain.

Lines 274-275: There seems to be and extra space or line break between the two lines.

Lines 323-324: Please add references.

Line 330: Questioned by whom? Please add references or clearly indicate if this is your own idea.

Line 351: What is QAC?

Line 378: Change “recalcitrant” to “resistance” or “resilience”

Lines 383-384: EPS is already discussed earlier. Please move the explanation to line 266.

Section 5. The focus should be more on the methodology, not results as indicated in the heading. Or otherwise consider renaming the section.

Lines 405-408: This paragraph is unnecessary here. Please remove or replace.

Line 433: Some of the text appears in yellow.

Conclusions: Future directions and challenges are discussed rather than conclusions.

Author Response

Review #1

This manuscript aims to review capabilities and survival mechanisms of subaerial biofilms in varying environmental conditions and stresses. First the effect of some mineral substrates on microbial colonisation are discussed, followed by a review of symbiotic communities in subaerial biofilms, responses to drought, radiation and chemical stress, and finally experimental and analytical methods. The focus is on (temporarily) dry conditions and UV radiation tolerance, although this is not very clearly stated in the manuscript. Instead, very broad definitions like “harsh”, “extreme” and “terrestrial” are, often misleadingly, used.

Following the Reviewer suggestions, the terms extreme and harsh conditions have been changed to dry and inhospitable throughout the MS including the title.

The rationale and aims of the review are valid, and recent literature is cited. However, the ambitious goals set in the abstract and introduction are not met. In many places the manuscript seems to be more like a list of summaries of previous studies rather than a proper review, which should compare, combine and make new connections between and conclusions from the existing data and studies. I suggest that the authors make the focus clearer and change the structure of the text such that the content and interpretations rather than details of individual studies are highlighted. Limitations of the methods and critical evaluation of the cited literature should also be included. More detailed comments are provided below.

Please see the replies below.

Major comments:

The title of the manuscript is misleading. Properties of mineral substrates are not discussed much, but instead the focus is on microbial communities. On the other hand, the latter part of the title alone is too broad considering the focus of the review. Please revise the title such that it describes the content of the manuscript.

The title has been revised according to the Reviewer’s suggestion.

In the abstract extreme and harsh conditions are mentioned without explanation. Please avoid too general terms and replace them by better defined terminology or explain what is meant here by extreme and harsh.

The terms extreme and harsh conditions have been changed to dry and inhospitable throughout the MS including the title.

Furthermore, I don’t see how climate change, extraterrestrial environments or system thinking mentioned in the abstract are considered and covered in the text, except in few separate sentences. Please make sure that the same topics are addressed in the abstract and the main text.

According to the Referee’s suggestion, the revised abstract does not include these issues.

There are many parts of the manuscript where specific terminology is used either loosely or without explanation. For example, terrestrial environments and lithosphere cover a wide range of different environments from deserts (both warm and cold) to aquifers (e.g. lines 82 and 86), lithosphere even including the oceanic crust. Thus, not all terrestrial or lithospheric environments are subaerial, dry or inhospitable. Again, better define and keep the focus and use the terminology accordingly.

The term lithosphere has been removed from the paper. Terrestrial environments have been changed to “dry/inhospitable terrestrial environments” throughout the MS.

For the sake of clarity, please also give short explanations for some specific terms such as chaperone gene and WspA (what are they doing, why are they important in this context?).

While editing the text, chaperone genes and WspA have been removed.

Especially in chapters 4 and 5 the review turns into a list of previous studies. The text could be condensed and it should report similarities and possible contrasts rather than detailed results of individual studies.

Following the Reviewer’s comment, the text has been condensed and similarities and possible contrasts added.

This is also reflected in the conclusions section of the manuscript, in which conclusions are not given, but the focus is solely on future directions/lack of data.

In agreement with the Reviewer’s comment, the conclusion section has been improved as follows:

(Lines 210-219): ‘SABs next-generation sequencing on mineral substrates are becoming increasingly available and shows the dominance of four main phyla, Cyanobacteria, Actinobacteria, Chloroflexi and Proteobacteria. Thus, SAB communities living in mineral environments benefit from goods exchanges and labor division in long-term partnerships between members of different groups.

Subaerial biofilms in inhospitable environments are multi-component open ecosystems sensitively tuned to the external environment, which provides not only nutrients, moisture and space, but also physical and chemical stressors that are drivers of biofilm formation, resistance and resilience. Despite the significance of model systems in SAB studies, works on this topic are rare, and most of them are based on mono-species biofilms.’

Detailed and minor comments:

Line 12: Minerals substrates alone cannot be considered as ecosystems. Please be more specific.

Lines 13-14: What is meant here by extreme conditions?

The text has been changed to “Despite being inhospitable ecosystems, mineral substrata exposed to air harbour…”

Line 17: Harshest conditions is a strong statement. What about microbes living in hot springs or inside nuclear reactors? Again, please be more specific what you mean by harsh.

The abstract has been condensed and now the words ‘Harshest conditions’ have been deleted.

Line 39: Mineral surfaces alone cannot be considered as specific environments, as water content, amount of organic matter and exposure to air can vary greatly between different environments containing mineral surfaces.

We added the adjective dry before terrestrial environment.

Line 44: Replace comma with “and” between desiccation and antimicrobial.

The change has been made.

Lines 46-47: Again the term harsh is used loosely. Also some marine habitats can be considered harsh. What part of the terrestrial biomass/biofilms is subaerial, i.e. more relevant to this review (also see the line 60, where low biomass is mentioned)?

The adjective harsh is no more present.

Line 55: Add “subaerial” before mineral substrates.      

The change has been made.

Lines 69-70: Please add reference(s) to the sentence “Indeed, most of the time…”

A reference has been added.

Lines 70-74: Consider moving this paragraph to section 2.

The sentence has been removed.

Lines 86-88: Please add reference(s), and also more clearly focus on subaerial biofilms; lithosphere is way too wide term to be used here.

References have been added.

Table 1: Layout of the table makes it difficult to read. For example it is not clear what references belong to what content.

The table has been modified for clarity.

Line 109: Replace “latest” by “least”.

The change has been made.

Lines 140-145: Reference(s) for this paragraph are missing.

A reference has been added.

Section 3.: Please consider reorganizing this section. It would be more logical to start with the classical example of lichens (from line 192 on) and only after that continue to the more recent examples and ideas. The section can also be shortened by condensing some of the more detailed content (same applies to section 4).

The exchange has been made and the section has been condensed.

Lines 219-220: This sentence should be rephrased, or maybe move “technically challenging” to the end?

The change has been made.

Lines 239-240: Please add reference(s).

The section has been condensed and the proper references made clearer.

Lines 244-246: This paragraph (probably from a previous review report?) should be removed.

The paragraph has been deleted.

Line 257: What studies? Please avoid using these, this etc. especially in the beginning of a paragraph.

The text of the revised paper has been corrected as suggested by the Reviewer.

Line 266: Abbreviation EPS is used for the first time, please explain.

We have added what the abbreviation stands for.

Lines 274-275: There seems to be and extra space or line break between the two lines.

The line break has been removed.

Lines 323-324: Please add references.

A reference has been added.

Line 330: Questioned by whom? Please add references or clearly indicate if this is your own idea.

The sentence has been changed to: “However, the main question we pose is how can we tell if…”

Line 351: What is QAC?

The acronym has been removed.

Line 378: Change “recalcitrant” to “resistance” or “resilience”

The change has been made.

Lines 383-384: EPS is already discussed earlier. Please move the explanation to line 266.

The change has been made.

Section 5. The focus should be more on the methodology, not results as indicated in the heading. Or otherwise consider renaming the section.

The heading has been changed.

Lines 405-408: This paragraph is unnecessary here. Please remove or replace.

The paragraph has been removed.

Line 433: Some of the text appears in yellow.

The change has been made.

Conclusions: Future directions and challenges are discussed rather than conclusions.

We have added a paragraph at the beginning of the section reporting the conclusion of the paper. The title of the section has been changed in “Conclusions and future directions”. 

Reviewer 2 Report

The manuscript submitted for review describes ecology of subaerial biofilms on air-exposed minerals. The work consists of 21 pages, contains three Figures, one Table and is based on 116 references. The authors divided the work into six chapters, including Introduction, Community assembly at the mineral / air interface, Biological interactions in SABs: a symbiotic playground, Stress resistance and resilience of SABs (physical stresses, chemical stresses), Lab-scale systems and mathematical models to study SABs, Conclusions.

               The subject of the review manuscript is very interesting and important, but in my opinion manuscript in this form is not suitable for publication. In my opinion, the most interesting chapters are Introduction and Conclusions, in which the authors point to the importance of SABs, including geochemical cycles, climate changes or extraterrestrial environments, and also which is very interesting, indicate the necessary directions of SABs research, including the creation of mathematical models and molecular metaanalysis. In the main part of the work we can read about the ecology of SABs including diversity, the coexistence of various trophic types, and adaptation to physical and chemical stress factors. These key chapters, although based on very good and recent literature, in my opinion do not constitute a complete review of knowledge on the subject, and even contain some ambiguities. The title indicates that the subject of the manuscript are subaerial biofilm on mineral substrates, while one of the main environment described in the overall manuscript is biological soil crust. I have two questions – (1) can we consider soil as air-exposed mineral? (2) can we consider SAB as biological soil crust or opposite? I also don’t understand the term "SAB inhabiting biological soil crust" (line 99, 105, 272). According to the literature biological soil crusts are the community of organisms living at the surface of soils, so biological soil crust can not be inhabited by SAB.

Author Response

Review #2

The manuscript submitted for review describes ecology of subaerial biofilms on air-exposed minerals. The work consists of 21 pages, contains three Figures, one Table and is based on 116 references. The authors divided the work into six chapters, including Introduction, Community assembly at the mineral / air interface, Biological interactions in SABs: a symbiotic playground, Stress resistance and resilience of SABs (physical stresses, chemical stresses), Lab-scale systems and mathematical models to study SABs, Conclusions.

The subject of the review manuscript is very interesting and important, but in my opinion manuscript in this form is not suitable for publication. In my opinion, the most interesting chapters are Introduction and Conclusions, in which the authors point to the importance of SABs, including geochemical cycles, climate changes or extraterrestrial environments, and also which is very interesting, indicate the necessary directions of SABs research, including the creation of mathematical models and molecular metaanalysis. In the main part of the work we can read about the ecology of SABs including diversity, the coexistence of various trophic types, and adaptation to physical and chemical stress factors. These key chapters, although based on very good and recent literature, in my opinion do not constitute a complete review of knowledge on the subject, and even contain some ambiguities.

The most important metagenomics studies conducted in these environments have been included in the new version of Table 1, together with the list of recent sequenced genomes of both prokaryotic and eukaryotic species isolated from biocrust, rocks and stone monuments. A paragraph about the quorum sensing role has also been added. Finally, works in these environments related to saline conditions are now reported.

The title indicates that the subject of the manuscript are subaerial biofilm on mineral substrates, while one of the main environment described in the overall manuscript is biological soil crust.

In line with the Reviewer’s comment, the title has been changed in ‘The ecology of subaerial biofilms in dry and inhospitable terrestrial environments’.

I have two questions – (1) can we consider soil as air-exposed mineral?

The answer is no and for this reason we have changed the title.

(2) can we consider SAB as biological soil crust or opposite?. I also don’t understand the term "SAB inhabiting biological soil crust" (line 99, 105, 272).

According to the referee’s suggestion, we have changed the text making clear that soil biological crusts are SABs as reported by Hoppert et al., 2004, Structure and Reactivity of a Biological Soil Crust from a Xeric Sandy Soil in Central Europe: ‘These subaeric biofilms are known as cryptogamic, microbiotic or biological soil crusts.’ For instance, the title of table 1 is now “Main characteristics of SABs as biological soil crusts or inhabiting natural (desert rocks) and built (stone heritage) environments.”

Reviewer 3 Report

The paper describes the ecology of SABs in harsh terrestrial environments.

It is an extensive and detailed review of the subject but perhaps it could be complemented with more information

I think it might be interesting to include a table citing the most important metagenomic studies conducted in these environments so that interested readers can access them.

On the other hand, no quorum sensing information is described and from my point of view should play an important role in these biofilm.

Is there no work related to these environments in saline conditions?

Another interesting aspect that could complement the work is to include data on prokaryotic species isolated from these extreme environments, some of them may be sequenced genomes that could provide much information.

The paper describes the ecology of SABs in harsh terrestrial environments.

It is an extensive and detailed review of the subject but perhaps it could be complemented with more information

I think it might be interesting to include a table citing the most important metagenomic studies conducted in these environments so that interested readers can access them.

On the other hand, no quorum sensing information is described and from my point of view should play an important role in these biofilm.

Is there no work related to these environments in saline conditions?

Another interesting aspect that could complement the work is to include data on prokaryotic species isolated from these extreme environments, some of them may be sequenced genomes that could provide much information.

Author Response

Reviewer #3

The paper describes the ecology of SABs in harsh terrestrial environments.

It is an extensive and detailed review of the subject but perhaps it could be complemented with more information.

I think it might be interesting to include a table citing the most important metagenomic studies conducted in these environments so that interested readers can access them.

As suggested by the Reviewer, the most important metagenomics studies conducted in these environments have been included in the new version of Table 1.

On the other hand, no quorum sensing information is described and from my point of view should play an important role in these biofilm.

Following the Reviewer’s suggestion, the text of the revised manuscript has been improved as follows:

(Lines 210-219) ‘The involvement of cooperative behavior may suggest a role for autoinducer-like compounds in such responses. Sharif et al. [47] reported that the epilithic cyanobacterium Gloeothece produce N-octanoyl-homoserine lactone as a signal molecule, altering gene expression in response in an autoinducer-like manner. Through microscopic investigations, Villa et al. [2, 45] revealed that the EPS matrix was not concentrated in only one single colony but extended along the mineral substrata, interconnecting cellular clusters together. Interestingly, it has been observed that the “calling distance” of quorum sensing can extend up to 78 μm between single species biofilms [48]. Thus, it is likely that communication and cooperation among segregated microcolonies can occur by diffusion of metabolites and QS molecules through the EPS network.’

Is there no work related to these environments in saline conditions?

In agreement with the Reviewer’s comment, the text of the revised manuscript has been improved as follows:

(Lines 270-273) ‘Interestingly, nitrification in the enrichments of some Negev desert samples was performed up to 400 mM NaCl (2.3% salt), a concentration close to that of seawater [65]. It is known that salt accumulation increases extracellular osmolarity as desiccation.’

(Lines 275-281) ‘Zhang et al. [66] investigated the importance of salinity along a natural salinity gradient in the Gurbantunggut Desert, Northwestern China. The researchers proved that microbial diversity was linearly reduced with salt accumulation, but community dissimilarity greatly increased with salinity differences. The latter counterintuitive finding has been explained by the fact that also unrelated taxa coexist in dry environments and the competitive exclusion of some closely related taxa.

Despite some manuscripts on salinity, the resistance and resilience of SABs to chemical stressors have been focused on the use of biocides in conservation treatments of stone monuments [67].’

Another interesting aspect that could complement the work is to include data on prokaryotic species isolated from these extreme environments, some of them may be sequenced genomes that could provide much information.

The list of recent sequenced genomes of both prokaryotic and eukaryotic species isolated from biocrust, rocks and stone monuments has been included in the new version of Table 1.

Round 2

Reviewer 3 Report

Once the corrections suggested in the previous version have been made, I believe that the work is suitable for publication

Author Response

The authors have responded to the reviewers’ comments and suggestions. Some minor corrections remain to do.

Define QS.

As requested by the Reviewer, the definition of quorum sensing has been provided in the revised version of the manuscript.

Lines 220-224: “The involvement of cooperative behavior may suggest the participation of quorum sensing signaling molecules in such responses. Quorum sensing (QS) is a cell-to-cell communication system depending on population density able to coordinate community behavior. Quorum sensing involves production of and response to diffusible or secreted signals called autoinducers.”

The comment of Reviewer 1 regarding Table 1 has not been adequately addressed. The table is most confusing. Specific pure cultures are listed in the last column, but how do they relate to the information in the other columns? The entries overlap in the columns; for example, bacteria (Bacillus spp.) in the last column match Crenarchaeota in the first column. The way this table has been set up makes no sense.

Table 1 has been revised.  The last column titled ‘sequenced genomes’ in Table 1, which was previously requested by the Reviewers, has been relocated to a new table of its own, Table 2. Furthermore, bullet points and borders have been added to Table 1 to improve readability.

A sentence to introduce the new Table 2 has been added in the revised text.

Lines 118-122: ‘Despite the sequencing efforts and advanced analytical tools, the complete genomes of microbial species from the topsoil, rock and human-made structures are scarce (Table 2). Genome sequences from cultivated and uncultivated microorganisms will allow deep investigations of the physiological traits that enable survival under inhospitable conditions, including the ability of these microorganisms to respond to future perturbations such as climate change and human impacts.’

The text has only one sentence regarding Table 1. The content of the long table deserves discussion, at least a paragraph, in the text.

Following the Reviewer’s comment, the text of the revised manuscript has been improved as follows:

Lines 85-89: ‘It is possible to draw parallels among key biofilm attributes that drive geomicrobial processes in the topsoil, rock and man-made structures in an inhospitable environment by comparing the main phylogenetic groups, functional traits, biologically-driven processes and drought-resistance mechanisms as reported in Table 1.’
